

# Characterising vertical turbulent dispersion by observing artificially released SO₂ puffs with UV cameras

Anna Solvejg Dinger[1,2], Kerstin Stebel[1], Massimo Cassiani[1], Hamidreza Ardeshiri[1], Cirilo Bernardo[3], Arve Kylling[1], Soon-Young Park[1,4], Ignacio Pisso[1], Norbert Schmidbauer[1], Jan Wasseng[1], and Andreas Stohl[1]

[1]NILU - Norwegian Institute for Air Research, NO-2007 Kjeller, Norway
[2]Institute of Environmental Physics, University of Heidelberg, D-69120 Heidelberg, Germany
[3]Aires Pty. Ltd., Mount Eliza, Vic 3930, Australia
[4]Center for Earth and Environmental Modeling Studies, Gwangju Institute of Science and Technology, Gwangju, Republic of Korea

**Correspondence:** Anna Solvejg Dinger (asd@nilu.no)

**Abstract.** In atmospheric tracer experiments, a substance is released into the turbulent atmospheric flow to study the dispersion parameters of the atmosphere. That can be done by observing the substance's concentration distribution downwind of the source. Past experiments have suffered from the fact that observations were only made at few discrete locations and/or at low time resolution. The COMTESSA project (Camera Observation and Modelling of 4D Tracer Dispersion in the Atmosphere) is

the first attempt at using ultraviolet (UV) camera observations to sample the three-dimensional (3D) concentration distribution in the atmospheric boundary layer at high spatial and temporal resolution. For this, during a three-week campaign in Norway in July 2017, sulfur dioxide (SO₂), a nearly passive tracer, was artificially released in continuous plumes and nearly-instantaneous puffs from a 9 m high tower. Column-integrated SO₂ concentrations were observed with six UV SO₂ cameras with sampling rates of several Hertz and a spatial resolution of a few centimetres. The atmospheric flow was characterised by eddy covariance

measurements of heat and momentum fluxes at the release mast and two additional towers. By measuring simultaneously with six UV cameras positioned in a half circle around the release point, we could collect a data set of spatially and temporally resolved tracer column densities from six different directions, allowing a tomographic reconstruction of the 3D concentration field. However, due to unfavourable cloudy conditions on all measurement days and their restrictive effect on the SO₂ camera technique, the presented data set is limited to case studies. In this paper, we present a feasibility study demonstrating that the

turbulent dispersion parameters can be retrieved from images of artificially released puffs. The 3D trajectories of the centre of mass of the puffs were reconstructed enabling both a direct determination of the centre of mass meandering and a scaling of the image pixel dimension to the position of the puff. The latter made it possible to retrieve the temporal evolution of the puff spread projected to the image plane. The puff spread is a direct measure of the relative dispersion process. Combining meandering and relative dispersion, the absolute dispersion could be retrieved. The turbulent dispersion in the vertical is then used to estimate

the effective source size, source time scale and the Lagrangian integral time. In principle, the Richardson-Obukhov constant of relative dispersion in the inertial subrange could be also obtained, but the observation time was not sufficiently long in comparison to the source time scale to allow an observation of this dispersion range. While the feasibility of the methodology



to measure turbulent dispersion could be demonstrated, a larger data set with a larger number of cloud-free puff releases and longer observation times of each puff will be recorded in future studies to give a solid estimate for the turbulent dispersion under a variety of stability conditions.

*Copyright statement.* TEXT

# 1 Introduction

A substance (a "passive scalar") injected into a turbulent atmospheric flow exhibits complex dynamical behaviour. Its distribution is chaotic, and the probability density function (PDF) of the scalar concentration field exhibits large fluctuations, which can depart substantially from Gaussian behaviour (Shraiman and Siggia, 2000). This behaviour can be difficult to capture with models. The direct numerical simulation of turbulence (Orszag and Patterson, 1972) is not feasible at Reynolds numbers typical for the atmospheric boundary layer (ABL), and other models used for tracer dispersion (e.g., Large Eddy Simulation or Lagrangian particle models) require parametrisations and/or validation based on atmospheric observations (e.g., Hanna, 1984; Arya, 1999).

Atmospheric tracer experiments are needed for constraining dispersion parameters. The first plume characterization experiments in the early 20th century were based on photographs of smoke clouds (Roberts, 1923; Nappo, 1981). More recent experiments released gaseous tracers such as sulfur dioxide ($SO_2$), sulfur hexafluoride or perfluorocarbons at one point and sampled concentrations in a network of ground stations (and sometimes by aircraft) downwind. The experiments carried out from the late 1950s to the early 1970s were the basis for many tools used in dispersion modelling (Pasquill, 1961; Gifford, 1961). As described in Hanna (2010), the Prairie Grass experiment (Barad, 1958), where near source (<1 km) dispersion of $SO_2$ was measured under many stability conditions was perhaps the one most useful for dispersion model validation. However, none of these experiments could capture the three-dimensional (3D) evolution of the dispersing plume in detail.

While the mean concentration is often well accessible to atmospheric measurements, fewer atmospheric observations are available for the higher PDF moments (variance, skewness, kurtosis). Yet, the higher moments are crucial if the relationship between the concentration fluctuations and their consequences is non-linear (Mylne and Mason, 1991). For instance, toxicity, flammability and odour detection depend on exceedances of concentration thresholds (e.g., Hilderman, 1999; Schauberger et al., 2012; Gant and Kelsey, 2012), and non-linear chemical reactions are influenced by tracer fluctuations if the reaction and turbulence time scales are similar (Brown and Bilger, 1996; de Arellano et al., 2004; Cassiani et al., 2013).

Atmospheric measurements of the concentration fluctuations in a dispersing plume have been performed by different groups and with different techniques (Mylne and Mason, 1991; Mylne, 1992; Yee et al., 1993, 1994). The most comprehensive observations were made with lidars measuring the backscattered signal from smoke particles (Jørgensen and Mikkelsen, 1993; Mikkelsen et al., 2002; Jørgensen et al., 2010). Other studies have used lidars to measure $SO_2$ concentrations (Schröter et al., 2003). A particular advantage of lidars is that they can measure concentrations throughout the ABL and not only near the





Earth's surface, where most in-situ measurements have been made. Nevertheless, even lidars provide only 1D measurements and, when scanning, cannot provide high time resolution in 2D or 3D. Thus, the 3D concentration distribution has never been measured at high time resolution.

The 3D concentration field is needed to evaluate the meandering and relative dispersion process in the three physical di-
rections. An important point to recognize is that the production and dissipation of concentration fluctuation for a dispersion scalar are intimately linked to the process of relative dispersion of puffs and the related process of center of mass meandering (Gifford, 1959; Csanady, 1973). Therefore parametrized expressions of relative dispersion are used in defining simplified models of concentration fluctuations (e.g. Luhar et al., 2000; Yee and Wilson, 2000; Cassiani and Giostra, 2002; Sawford, 2004; Cassiani et al., 2005; Marro et al., 2015, 2018).

One possibility to indirectly measure 3D tracer concentrations at high time and space resolution (thus able to capturing concentration fluctuations) are ultraviolet (UV) cameras. These cameras can measure sulfur dioxide ($SO_2$) column concentrations with a sampling frequency of several Hertz (Kern et al., 2010; Lübcke et al., 2013). Clouds in the image background can cause inhomogeneous illumination of the sky, which complicates the $SO_2$ column concentration retrieval of $SO_2$ camera images. While efforts have been made to correct cloud effects (Osorio et al., 2017), it is generally recommended to measure during
clear-sky conditions (Kern et al., 2010; Kantzas et al., 2010). To date, $SO_2$ cameras have been used mostly to monitor $SO_2$ emissions from volcanoes (Burton et al., 2015), power plants (McElhoe and Conner, 1986; Smekens et al., 2015) and ships (Prata, 2014). While each individual camera measures only 2D distributions of $SO_2$ column concentrations, a combination of several such cameras should allow a tomographic reconstruction of the 3D $SO_2$ distribution.

However, to our knowledge such a tomographic setup has never been used successfully. The COMTESSA project (Camera
Observation and Modelling of 4D Tracer Dispersion in the Atmosphere) is the first attempt at using camera observations to study tracer dispersion in the ABL. For this, we artificially release $SO_2$ into the atmosphere and observe its dispersion with UV cameras.

In this paper, we present results from the first COMTESSA field campaign, which was conducted to test our new instrumentation. Not all equipment was fully operational yet, but we were nevertheless able to collect a valuable data set using six UV
cameras and meteorological instrumentation. Here, we first describe the release experiments (Sect. 2) and how a tomographic setup of UV cameras can be used to quantify the dispersion of artificially released $SO_2$ puffs in the ABL (Sect. 3 and 4). As an example, the 3D trajectories and spreads of six puffs within a short time interval of 60 s are reconstructed (Sect. 5). Then, the time evolution of puff meandering, relative and absolute dispersion are retrieved enabling estimations of turbulent time scales (Sect. 6). The data set does not contain a sufficiently large number of puffs for a reliable statistical analysis, however,
the feasibility of the method is demonstrated.

## 2   Artificial release experiment

The first COMTESSA campaign was performed at a military training ground (11.5°E, 61.4° N) about 28 km northeast of the small city of Rena, Norway, from 3 to 21 July 2017. The experimental site is located in a remote forested mountain area





at an altitude of 850 m above sea level. It is a fenced-in flat gravel field with dimensions of about 900 m × 400 m, which is normally used for ammunition testing by the Norwegian military. Three nine metres high masts equipped with eddy covariance measurement systems were set up to measure the turbulent fluxes of heat and momentum. From the top of one of the masts, pure $SO_2$ gas was released, piped from $SO_2$ bottles at the ground using a commercial blower. The blower speed was set such

that the release was nearly isokinetic. That was achieved by adjusting the flow in the pipe to the wind speed monitored online with a sonic anemometer at source elevation. The pipe had a diameter of 12.5 cm at the release point. Fig. 1 shows a picture of the top of the release mast.

The weather conditions were generally not favourable for our experiment, with several cyclones passing over Fennoscandia during the campaign period. Daily average temperatures at a meteorological station located in the immediate vicinity (Rena

øvingsfelt) ranged between 6.8 and 11.7° C, except for the last two days when they rose above 13° C. On 13 of the 19 campaign days, precipitation was recorded, and winds were often strong (up to 9 m s$^{-1}$). Conditions were suitable for instrument testing on several days, but clear-sky conditions were rare. The best conditions were encountered on 20 July when a ridge of high pressure built over Southern Fennoscandia. While even on that day there was no period when the sky was entirely free of clouds, there were periods with relatively little cloud cover, enabling clear-sky camera observations for some viewing directions

and yielding clouded scenes for the other cameras. In this paper, we will therefore present results only for this day.

On 20 July, $SO_2$ was released during several experiments, including both several continuous plumes (between 7:19 and 9:53) and nearly-instantaneous puffs (between 10:24 and 10:47). Six identical UV $SO_2$ cameras observed the $SO_2$ releases, resulting in column-integrated $SO_2$ concentration images from six directions. The six $SO_2$ cameras observed an overlapping volume of roughly 40 m × 40 m × 20 m, centred circa 18 m downwind of the release point. The cameras were arranged on the ground

in a half circle with a radius of 160 m around this volume. The release point is visible in the field of view of every camera. Additionally, a meteorological tower, located a few hundred meters northwest of the release tower, is visible in the field of view of some cameras. A map of the setup is shown in Fig. 2 and detailed quantitative information can be found in appendix A1.

The $SO_2$ cameras were custom-built for the COMTESSA project (Fig. 3). At the core of each $SO_2$ camera are two UV cameras from PCO (pco.ultraviolet), which record images at two different wavelengths. The wavelengths are selected by mounting two

Asahi Spectra band-pass filters (10 nm bandwidth) at 310 nm and 330 nm, respectively, in front of the cameras. The filters are mounted between the CCD sensor and a 25 mm quartz lens from Universe Kogaku. This setup attenuates radial sensitivity changes due to different light paths through the filter for off-axis rays compared to mounting the filters in front of the lens (Kern et al., 2010). The cameras' CCD sensors have $N_i = 1392$ pixel columns and $N_j = 1040$ pixel rows, resulting in a image resolution of a few centimetres at object distances of a few hundred meters. The camera properties are summarised in Table 1.

During the experiment, the exposure times were chosen manually such that the 14-bit-sensor was roughly 80% saturated. On 20 July, the exposure times for the 310 nm-camera were between 160 ms and 200 ms at apertures of f/2.8. Further, each camera contains an AvaSpec-ULS2048x64 spectrometer from Avantes for robust $SO_2$-calibration. The spectrometer is coupled via a 3x200 $\mu$m cross section converter fibre from loptek to a telescope, pointing in the same direction as the UV cameras. The telescope consists of a quartz lens from Thorlabs with 100 mm focal length and a Hoya U-330 filter which prevents stray light

to enter the detector. This setup results in a telescope field of view of 0.572° which corresponds to a disk with a 52-pixel




diameter within the UV camera image. In the future, a built-in GPS will be used to obtain accurate space and time information. However, during the experiment in summer of 2017, the GPS data were not yet recorded and, therefore, the individual $SO_2$ cameras were synchronised in time by tracking of distinct $SO_2$ features after the experiment (see appendix A2 for details).

Meteorological measurements were collected on the release tower at three vertical positions (2 m, 5.4 m and 8.7 m) using a

state-of-the-art measuring system from Campbell Scientific. It included sonic anemometers at all three levels (model CSAT3A and CSAT3B respectively) measuring three wind components and sonic temperature with 50 Hz sampling frequency. Additionally, an EC150 gas analyser was coupled to the lowest level. It measured simultaneously water vapour and carbon dioxide densities at 50 Hz, as well as the atmospheric pressure and temperature at lower frequency. During the puff release experiment on 20 July, the mean wind velocity at the source was 5.22 m s$^{-1}$ and the fluctuations of the vertical velocity component were

$\sigma_w^2$=0.283 m$^2$ s$^{-2}$. The derived value of the Obukhov length $L$=-6.22 indicates an unstable atmosphere with convective conditions. Further measured and derived parameters are summarised in Table 2 and the applied post-processing of the wind data is detailed in Appendix A3.

## 3 Turbulent dispersion

### 3.1 Description of turbulent dispersion

The absolute dispersion $\sigma_i^2$ describes the spread of a scalar relative to a fixed origin along the coordinate axis $i$. Mathematically, $\sigma_i^2$ is the variance of the 1D mean concentration distribution along the considered axis. Taylor (1921) demonstrated that the absolute dispersion is directly linked to the Lagrangian autocorrelation function of the motion of one particle. According to Taylor's theory and assuming homogeneity and an exponential autocorrelation function (see e.g. Arya, 1999) the evolution of the absolute dispersion with time $t$ in the vertical coordinate $z$ is modelled as

$$\sigma_z^2(t) = 2 \cdot \overline{w'^2} \cdot T_L^2 \left( \frac{t}{T_L} - \left[ 1 - \exp(-\frac{t}{T_L}) \right] \right) \tag{1}$$

with the vertical velocity $w(t)$ and the vertical Lagrangian time scale $T_L$. Assuming homogeneity, the variance of the vertical velocity $\sigma_w^2 = \overline{w'^2}$ can be obtained from the velocity monitored by a sonic anemometer placed at the source location. Given the very short range of our current measurements this is an acceptable approximation. The Lagrangian time scale $T_L$ cannot be measured directly by a fixed point measurement, instead the Eulerian time scale $T_E$ can be obtained from such measure-

ments. Hay and Pasquill (1959) assumed that the Lagrangian and Eulerian time scales have a fixed ratio $\beta = T_L/T_E$. The proportionality constant $\beta$ can be found using the relationship proposed by Hanna (1981),

$$\beta i \approx 0.7 \tag{2}$$

where $i = \frac{\sigma_w}{\overline{u}}$ is the turbulence intensity in the along wind direction with mean velocity $\overline{u}$.

The absolute dispersion of an ensemble of puffs (or clusters of particles) can be assumed to be partitioned between two

statistically independent components: the meandering of the puffs as a whole with respect to the source location, and the spread of the puffs around their centre of mass, called relative dispersion. This is sketched in Fig. 4. In mathematical terms, the




variance of the mean concentration distribution $\sigma_i^2$ is decomposed as a sum of the variance of the centre of mass distribution $\sigma_{m,i}^2$ and the variance of the concentration of the puff relative to its centre of mass $\sigma_{r,i}^2$,

$$\sigma_i^2(t) = \sigma_{m,i}^2(t) + \sigma_{r,i}^2(t) \tag{3}$$

Experimentally, the variances are obtained by averaging over multiple realisations of single puffs.

A cluster of particles released at the same time from a finite source will follow slightly different paths and form a distribution around its centre of mass. The relative dispersion is therefore influenced by the source size $r_0$, i.e. the initial separation of the particles. Batchelor (1952) showed that the initial spread of a puff, or cluster of particles, is initially governed by

$$\langle \mathbf{r}^2 \rangle = \tfrac{11}{3} C_k \epsilon^{2/3} r_0^{2/3} \cdot t^2 + r_0^2 \quad \text{for } t \ll t_s \tag{4}$$

where $C_k$ is the Kolmogorov's constant for the longitudinal structure function in the inertial subrange, $\epsilon$ is the mean dissipation

of turbulent kinetic energy and $t_s = (r_0^2/\epsilon)^{1/3}$ is the characteristic time of the source based on inertial range scaling. Here, $\mathbf{r}$ is the 3D separation between two particles of the cluster and $\langle \mathbf{r}^2 \rangle$ is the ensemble mean square separation between all particles of the cluster. In homogeneous isotropic turbulence, $\langle \mathbf{r}^2 \rangle$ is related to the 1D relative dispersion as $\sigma_{r,i}^2 = \langle \mathbf{r}^2 \rangle/6$ (see e.g. Franzese and Cassiani, 2007). Equation (4) reduced to the vertical component reads then

$$\sigma_{r,z}^2(t) = \sigma_{r,z_0}^2 + 6^{-2/3} \cdot \tfrac{11}{3} C_k \epsilon^{2/3} \sigma_{r,z_0}^{2/3} t^2 \tag{5}$$

with the 1D initial vertical separation $\sigma_{r,z_0}^2 = r_0^2/6$

For larger times $t \gg t_s$, i.e. for the inertial range, the spread of the puff is proportional to the Richardson-Obukhov constant $C_r$ according to the Richardson-Obukhov scaling (e.g. Monin and Yaglom, 1975)

$$\langle \mathbf{r}^2 \rangle = C_r \cdot \epsilon \cdot t^3 + r_0^2 \quad \text{for } t \gg t_s \tag{6}$$

The value of the Richardson-Obukhov constant is uncertain, as it is difficult to estimate from experiments and numerical sim-

ulations (see Franzese and Cassiani, 2007, for a detailed discussion). However, $C_r$ and the directly related relative dispersion are important for models as the relative dispersion defines the effective rate of mixing of a puff and therefore the decay rate of concentration fluctuations (e.g. Sawford, 2004; Cassiani et al., 2005; Pinsky et al., 2016; Marro et al., 2018).

### 3.2  Turbulent dispersion from image data

Videos of column-integrated concentrations (CIC) of an instantaneous release of a passive tracer can be used to measure

different aspects of turbulent dispersion, especially when simultaneous images from different directions are available. The CIC images contain direct information about the puffs' position and spread projected to the image plane (see Fig. 5 for a sketch and Fig. 6 for an example image). The image plane is spanned by two discrete coordinate axes $i = [1, .., N_i]$ and $j = [1, .., N_j]$ describing the image columns and rows. We define a rectangular extension of the projected puff, the so-called region-of-interest (ROI), to distinguish different puffs that may be present in the image, and to reduce the impact of noise. Then, the total signal



$S_{tot}$ of the puff (or, in statistical terms, the zeroth moment of the column-integrated concentration PDF) is given by

$$S_{tot} = \sum_{i,j \text{ in ROI}} S(i,j) \tag{7}$$

where $S(i,j)$ is the CIC at pixel $(i,j)$. The centre of mass (CM) of the puff in the image plane (first moment of the column-integrated concentration PDF) is given by

$$\begin{pmatrix} i_{cm} \\ j_{cm} \end{pmatrix} = \frac{1}{S_{tot}} \sum_{i,j \text{ in ROI}} S(i,j) \begin{pmatrix} i \\ j \end{pmatrix} \tag{8}$$

The spread of mass around its centre as given by the variance (2nd moment of the column-integrated concentration PDF) is described by the weighted covariance matrix $\mathbf{C}$. The diagonal elements of $\mathbf{C}$ are the spreads of the $SO_2$ puff in the image plane along the image columns and rows respectively. Accordingly, the horizontal spread along pixel columns is given by

$$C_{1,1} = \frac{1}{S_{tot}} \sum_{i,j \text{ in ROI}} (i - i_{cm})^2 \cdot S(i,j) \tag{9}$$

$$= \frac{1}{S_{tot}} \sum_{i,j \text{ in ROI}} i^2 \cdot S(i,j) - i_{cm}^2 \tag{10}$$

The spread along pixel rows $C_{2,2}$ is calculated equivalently.

However, retrieving quantitative dispersion parameters such as the total mass from the camera images requires that the pixel dimensions in the virtual object plane, containing the puff, are known. A pixel is, strictly speaking, a solid angle defined by the focal length of the camera lens $f$. Thus, for knowing the apparent width of the pixel at the position of the puff, the distance $d$ of the $SO_2$ puff to the camera needs to be known. Then, the apparent width of a pixel $s_p(d)$ is given by

$$s_p(d) = s_i \cdot \frac{d - f}{f} \tag{11}$$

where $s_i$ is the physical width of the pixel on the CCD sensor. The height of the pixel in the virtual object plane is calculated analogously and it is equal in case of a sensor with square pixels. In the following, square pixel are assumed for simplicity.

When the puff's 3D extension is small in comparison to the distance from the puff's CM to the camera, differences in distance over the puff's extension can be neglected and a constant scaling can be assumed for the whole ROI. Scaling the CIC images with the pixels' apparent area $s_p^2$, relates the image to a global reference system. It follows for the total mass $M$ of a puff

$$M = s_p^2(d) \cdot S_{tot} \tag{12}$$

and the horizontal puff spread in square meters

$$C_{1,1} \, [\text{m}^2] = s_p^2(d) \cdot C_{1,1} \, [\text{pixel}^2] \tag{13}$$

The spread describes the mass distribution relative to the centre of mass and projected to the image plane. It is hence connected to the relative dispersion. Depending on the relative orientation of the mean wind direction and the camera's optical



axis, it can equal the vertical, along- or across-wind direction in some cases. In other cases, assumptions of the plume shape have to be made (e.g. Gaussian plume) or a 3D reconstruction of the distribution is necessary. When detecting the puff's CM with more than one camera, the CM position in a global coordinate system can be reconstructed. For analysing the statistical nature of the turbulent dispersion, an ensemble of puff releases is required. Then, the meandering is calculated from the variance

of the 3D CM positions and the relative dispersion is connected to the measured puff spread.

## 4   Retrieval of CM trajectories and spread of artificially released puffs using a tomographic setup of SO$_2$ cameras

In this study, SO$_2$ CIC images recorded simultaneously with six UV SO$_2$ cameras are the basis for the retrieval of puff spreads. An example of such an image can be seen in Fig. 6 and the imaging technique will be described in the following Sect. 4.1. The puffs are detected automatically within the image using common image processing techniques (Sect. 4.2). This allows for

calculating the CM and spread of the puff projected to the image plane. Making use of the tomographic setup of six cameras (see Fig. 2 and Fig. 5) and the previously measured, projected CMs, the 3D trajectories are reconstructed (Sect. 4.3). The 3D trajectories then allow for scaling the measured puff spreads to square meters.

### 4.1   SO$_2$ camera imaging technique

The SO$_2$ camera method (Mori and Burton, 2006) is based on the principle of absorption spectroscopy of backscattered

sunlight. Gaseous SO$_2$ molecules exhibit a distinct, wavelength-dependent absorption cross section in the ultraviolet $\sigma(\lambda)$, where $\lambda$ is the wavelength. The relationship between the light intensity before and after passing through a SO$_2$ cloud - $I_0(\lambda)$ and $I(\lambda)$ - is described by the Beer-Lambert law

$$I(\lambda, L) = I_0(\lambda) \cdot \exp\left(-\int_0^L \sigma(\lambda) \cdot c(l)\,\mathrm{d}l\right) \tag{14}$$

$$= I_0(\lambda) \cdot \exp\left(-\sigma(\lambda) \cdot S\right) \tag{15}$$

where $c(l)$ is the SO$_2$ concentration at position $l$ along the light path $l \in [0, L]$ through the SO$_2$ cloud and $S = \int_0^L c(l)dl$ is the SO$_2$ slant column density (SCD) along this light path. Generally, radiative transfer effects (e.g multiple-scattering inside the SO$_2$ cloud and light dilution (Kern et al., 2013; Campion et al., 2015)) have to be taken into account when translating the slant column density to the column-integrated concentration. However for this study, the effects are negligibly small due to the absence of aerosol and the small extension and short distance of the SO$_2$ puffs to the cameras. Therefore, the slant column

densities correspond nearly exactly to the column-integrated concentrations and are used as such throughout the publication.

The SO$_2$ cameras record intensity images of the SO$_2$ cloud $I(\lambda)$. Images of the clear sky intensity $I_0(\lambda)$ can be measured in the same direction when the SO$_2$ cloud is not present (i.e. before or after a release experiment). The SO$_2$ slant column density $S$ is proportional to the optical density $\tau(\lambda)$, which is retrieved from the two images by

$$\tau(\lambda) = -\ln \frac{I(\lambda)}{I_0(\lambda)} = \sigma(\lambda) \cdot S \tag{16}$$





Using a narrow bandpass filter in the ultraviolet (typically 310 nm), a narrow spectral band of strong $SO_2$ absorption is selected. While high-precision laboratory measurements of the $SO_2$ absorption cross section $\sigma(\lambda)$ are available (e.g. Vandaele et al., 2009), calibration from $\tau(\lambda)$ to $S$ is nevertheless necessary due to uncertainties of the exact filter function. The measured optical density images $\tau$ are approximated to $SO_2$ SCDs by linear regression using absolute measurements of the SCDs

$$S = a\tau + b \tag{17}$$

$$= -a\ln\frac{I(\lambda)}{I_0(\lambda)} + b \tag{18}$$

where $a$ and $b$ are calibration constants. Such measurements are available from images of gas cells containing a known amount of $SO_2$ and/or from spectra of a built-in spectrometer (Lübcke et al., 2013). Making use of the differential optical absorption spectroscopy (DOAS) technique, a time series of precise point measurements of the $SO_2$ SCD corresponding to a small pixel area within the camera images can be retrieved and correlated to the image time series.

Moving, meteorological clouds behind the $SO_2$ cloud can change the illumination of backscattered sunlight between the two images $I_0(\lambda)$ and $I(\lambda)$. This leads to artefacts in the retrieved SCD images which can be of the same magnitude as the $SO_2$ signal. While $SO_2$ camera measurements under cloudy conditions should therefore be avoided if possible, we could obtain only such measurements due to the weather conditions during the experiments.

In this publication, the background images $I_0$ were taken from the same direction between two puff releases and the images were calibrated using the built-in spectrometer. Note that, contrary to typical applications (e.g Mori and Burton, 2006; Kern et al., 2015), measurements at only one wavelength ($\lambda$=310 nm) can be used for the analysis due to the absence of broadband absorption from additional aerosol in the $SO_2$ cloud. More details on the retrieval steps used in this publication can be found in the Appendix B1.

## 4.2 Detection of individual puffs in image plane

The position and spread of individual $SO_2$ puffs are tracked from the release point automatically. For that, rectangular ROIs containing the full puff need to be detected. Such a detection can be difficult for several reasons. (1) The images contain partly up to two puffs and artefacts from clouds, which can imitate $SO_2$ absorption. (2) Small fractions of the puffs can separate completely from the puffs. (3) The images are noisy, making correct identification of pixels with low $SO_2$ values at the edges of the puffs difficult. In consequence, the ROI has to be large enough to contain the full puff but small enough to exclude additional puffs and clouds.

To overcome these challenges, we choose an approach combining iterative tracking from the release point and applying signal thresholds to two noise-reduced versions of the original image. In this way, the ROI could be detected robustly and the total signal, CM and spread of the puff could be retrieved from the original image. Details of the detection algorithm can be found in Appendix B2. Further, this approach allows to track several puffs in the same image frame as long as they are separable. Single clouds can be ignored if they are not at the same position as the puffs and even the position of a puff in front of an overcast sky can be constrained spatially, even if not fully detected.





### 4.3 3D trajectories and pixel scaling

The previously retrieved CMs projected to the image planes of the cameras can be used to retrieve the 3D trajectories of the CM in the global coordinate system. These allow for calculating the distances between a puff and the individual cameras at any given time. Subsequently, the scaling factor (Eq. (11)) for the other moments of the PDF (Eq. (12) and (13)) can be determined.

5    The individual images of the six cameras are recorded at irregular time intervals due to differences in exposure and read-out times. Combining the irregular image times, the derived 3D trajectories in the global coordinate system were retrieved on an arbitrary-chosen discrete, regular time grid. Here, 250 ms was chosen so that at least one image of every camera lies within each interval. The time series of the CM image coordinates of the six cameras are synchronised and interpolated to this common time grid.

10    For every time step and each camera, the line-of-sight line from the position of the camera through the detected CM in the image plane at $(i_{cm}, j_{cm})$ is determined by calculating the azimuth and elevation angle. The azimuth angle of the CM is the sum of the camera's azimuth angle $\alpha$ of the optical axis and the relative azimuth angle of the CM to the optical axis

$$\alpha_{cm} = \alpha + \arctan\left(\frac{(i_{cm} - i_c) \cdot s_i}{f}\right) \tag{19}$$

where $i_{cm}$ is the pixel column of the CM, $i_c = \frac{N_i}{2}$ is the pixel column containing the optical axis (approximated by the central 15 pixel), $s_i$ is the physical pixel width on the CCD sensor and $f$ is the focal length. The elevation angle is calculated analogously based on the camera's elevation angle and the pixel row $j_{cm}$ of the CM.

At every time step, the position of the CM in the global reference system is then calculated based on the line-of-sight lines of all available cameras using a least squares optimisation: the CM is the point in the global reference system which minimises the square distance to all lines. The CM can be calculated for every time step for which data from at least two cameras was 20 available. However, in this analysis data from at least three cameras were used in order to reduce discontinuities caused by uncertainties in the cameras' position and pose. The reconstructed 3D trajectories can then be used to determine the distances between the cameras and the puffs at any given, individual image time.

## 5 Results

On July 20 between 10:24 and 10:47 UTC, a total of 140 puffs were released almost instantaneously, each puff containing 25 between 0.8-1.2 g of $SO_2$. The differences in mass originate from the manual opening and closing of the release valve. Due to the changing cloud cover, the analysis of the $SO_2$ camera images requires that background images are selected manually every 30 to 40 s of data. Additionally, puffs overlapping with clouds or each other limit the analysis further. Hence, for this feasibility study, results for a continuous 1-minute interval (10:29:50 to 10:30:50 UTC), containing six subsequent puffs are presented.

The six puffs can be tracked with all cameras in the image plane (Fig. 7) and the 3D CM trajectories can be reconstructed 30 successfully over up to 58 m (Fig. 8- 10). Typical distance/extension ratios are around 100, justifying the assumption of constant scaling throughout the ROI. The puffs move in two dominant directions (approx. $0\,°$ and $30\,°$) in good agreement with the



overall measured wind direction. Figure 11 displays the evolution of the moments of the spatial distribution (total mass, horizontal and vertical spread) of the six puffs. These are discussed in more detail in the following.

## 5.1 Total SO$_2$ mass

The total SO$_2$ mass of the puffs is conserved, since loss mechanisms (e.g dry deposition and oxidation of SO$_2$) can be neglected
on such short time scales as the ones observed. A change of the measured absolute mass and differences between the signals of different cameras are indications for measurement biases and limitations. These include besides others the cameras' detection limits, incomplete detection of the puff by the derived ROI, additional signals (both negative and positive) from cloud artefacts, uncertainties in the trajectory retrieval and thus the scaling parameter, and radiative transfer effects.

The upper panel in Fig. 11 shows the total mass of the puffs as observed by four of the six cameras. Cameras 5 and 6 were
excluded due to overly pronounced additional signals from the cloudy sky. The background images including the cloud cover for each camera were optimised for the time of the second displayed puff (indicated by shaded area). For this puff, the retrieved total SO$_2$ masses from the four cameras show good agreement: The mass first increases to circa 1.2 g SO$_2$ while the puff is released and then stays constant for all cameras until the puff is no longer tracked. For the other puffs - and thus increasing time difference to the background images - the relative differences between the cameras increases (up to 50%).

The total mass is strongly affected by clouds, which add both negative and positive signals to the total mass. For camera 3, single clouds are visible along the full pathway of the puff, resulting in a generally overestimated signal. For cameras 2 and 4, single clouds appear only from the middle of the image. Thus in this case an underestimation of the total mass starts only a few seconds after the release. Cameras 5 and 6 (not shown), however, fail to reproduce the released total mass even for the second puff. Camera 1 observes the puffs free of additional signal from clouds and hence catches the correct mass. However, due to
it's frontal alignment to the puff's propagation direction, subsequent puffs might overlap. This was the case for the three puffs between 10:30:20 and 10:30:50. For these puffs no separate mass or spread information can be extracted.

As the mass cannot be retrieved accurately for all data points, it can be assumed that the puff spread would be affected in a similar way by the additional signal due to clouds or overlapping puffs. Therefore such data points should be discarded from the analysis of the turbulent dispersion. Only measurements for which the total mass lies within a physically reasonable range
(here, 1.0 to 1.3 g s$^{-1}$) are included for further discussion.

## 5.2 Puff spread

Figure 11 shows the puff spread (Eq. (10) and (13)) in the image plane for four cameras. It is pointed out that these puff spreads are projected to the camera's object plane at the position of the puff. Hence only the puff spread perpendicular to the camera's optical axis is measured (see Fig. 5).

In the horizontal, the cameras' relative orientation lead to different projections and thus not directly comparable puff spreads. Camera 1 views the puffs almost frontal and thus the retrieved puff spreads are across-wind in first approximation. Cameras 2 and 3 view the puffs nearly perpendicular to their propagation direction, hence they measure approximately the along-wind



spread and their results agree reasonably well. The limited comparability of the cameras and the short data set of only six puffs does not allow for a further analysis in terms of horizontal dispersion.

The elevation angles of the cameras are comparably small (2.3° - 3.9°). The vertical projection to the image plane is negligibly small for these elevation angles (cos(3.9°) = 0.998). Hence the measured vertical puff spreads correspond to the real

vertical spread of the puff and thus are comparable between the cameras. The measured values of the four cameras agree with each other. In the following discussion only the vertical puff spread is considered for simplicity.

## 6   Turbulent dispersion in the vertical

For the analysis of the turbulent dispersion it would be necessary to observe a large number of instantaneous releases under stationary atmospheric conditions. For this study, only six subsequent puffs were selected due to the limitations of the mea-

surements under cloudy conditions. The total analysed time span is 60 seconds. Hence, the following discussion of the results should be considered as a demonstration of method rather than a robust estimate for parametrisation of turbulent dispersion.

### 6.1   Meandering

The vertical meandering $\sigma_{m,z}^2$ was calculated as the variance of the ensemble average of the CM trajectories. The shortest trajectory of the six puffs extended over 8 s after release. The ensemble average was calculated for every time step up to this

time in order to give a constant weight to all detected trajectories (i.e. at every point in time, the same number of trajectories is averaged). Figure 12 shows the meandering for the six puffs and, additionally, it shows the meandering when additional puff trajectories from the full duration of the experiment are included. This enabled an assessment of the uncertainty of the meandering estimate. The number of included trajectories was varied by simultaneously reducing the minimum trajectory length and increasing the time interval. Including a different number of puffs can lead to both a higher and lower $\sigma_{m,z}^2$. The

meandering is generally larger when more trajectories are included, particularly in the first few seconds all values lie above the meandering for the six puffs only. The meandering calculated from the full time period at medium trajectory lengths (7 s) was up to two times higher. The increase might originate from atmospheric variability or from the poor statistics. Additionally, a decreasing trend with increasing minimum trajectory length can be observed. This might be explained by the experimental setup. Some trajectories could get discarded during the data processing due to e.g. clouds in the background or the puff moving

out of the field of view. This leads to an effective data reduction to only certain directions and therefore an underestimation of the vertical meandering. In conclusion, the meandering shows a high dependence on the included trajectories, which can be only resolved if a higher number of puffs is available.

### 6.2   Relative dispersion

The relative dispersion is the spread of the $SO_2$ distribution around its centre of mass. It can therefore be estimated for each

individual puff. The spread of the six puffs, averaged over cameras 1-4, and their ensemble average are plotted on a double logarithmic-scale in Fig. 13. The observation time length is too short to observe a clear transition from the $t^2$ to the $t^3$ regime.





The slope suggests that only the initial $t^2$-regime, in which the relative dispersion is dominated by the source size, is observed. The effective vertical source size can then be estimated from the well defined $t^2$ expansion regime, i.e. the linear part of the temporal evolution of the ensemble average $\sigma_r^2(t)$ with a slope of two in the log-log plot. Following Eq. (5), the resulting vertical source size is fitted to $\sigma_{r,z_0}$=8.3 cm and compares to the radius of the release outlet (6.25 cm). The increased number

can be explained by the jet created at the source by the blower. Assuming an isotropic source, the source time scale was estimated to $t_s = (6 \cdot \sigma_{r,z_0}^2/\epsilon)^{1/3}$=2.6 s from the vertical source size and energy dissipation rate $\epsilon$. The resulting time lies in the middle of the observed time period, making it possible to theoretically observe the onset of the transition to the inertial subrange but failing however to characterise the relative dispersion in the inertial subrange. Longer observation times would enable an estimate of the Richardson-Obukhov constant by fitting Eq. (4) to the data using the measured value for the energy

dissipation.

### 6.3 Absolute dispersion

The absolute dispersion describes the spreading of particles relative to a fixed origin. It is calculated as the sum of meandering and relative dispersion (Eq.( 3)). In Fig. 14 the 1D absolute dispersion in the vertical dimension is displayed. The figure contains two parametrisations ($\overline{w'^2}$, $T_L$) of Taylor's theorem (Eq. (1). In both cases, $\overline{w'^2}$ is taken from the sonic anemometer

data close to the source. The estimate of the Lagrangian time scale differs: $T_L$ is either modelled from the measured Eulerian time scale $T_{E,z}$=3.07 s from the same anemometer data using the empirical constant $\beta$=6.87 (Eq. (2)) or fitted to the absolute dispersion retrieved from the image data. The modelled Lagrangian time is $T_{L,z}^{model}$=21.1 s and the fitted one is $T_{L,z}^{fit}$=5.9 s. The fitted Lagrangian time scale relates to the measured Eulerian time scale with $\beta^{fit} = \frac{T_{L,z}}{T_{E,z}} = 1.9$ and lies within the previously reported range of 1 to 10 (see e.g. Hay and Pasquill, 1959; Arya, 1999).

Here we report the absolute dispersion during the first 8 seconds after the release. Hence, all measurements were recorded at times below both the modelled and the measured Lagrangian times scales. For times much smaller than the true Lagrangian time scale, the absolute dispersion can be approximated by a quadratic relation, $\sigma_z^2 \approx \overline{\sigma_w^2} t^2$, independent of the Lagrangian time scale, hence making an estimation of the latter nearly impossible for short observation times. Therefore, even if a retrieval of the Lagrangian time scale from the current image data is possible, it is not reliable since the puff observation time does not

exceed the Lagrangian time scale.

Further, the absolute dispersion was observed close to the source when it is dominated by the meandering ($\sigma_m^2 \approx 10\sigma_r^2$). The absolute dispersion has therefore an uncertainty similar to that of the meandering (see Sect. 6.1 above).

### 7 Conclusions and future work

During the first COMTESSA experiment, the passive tracer SO$_2$ was released in the ABL to study its dispersion based on images

from six UV SO$_2$ cameras. The absolute dispersion, as well as the relative dispersion and meandering of an ensemble of six puffs could be retrieved by performing a reconstruction of the 3D trajectories of the centre of mass positions of instantaneous





puff releases. The measured absolute dispersion understates both the modelled and fitted parametrisations of Taylor's theorem due to the underestimation of the puff meandering.

We showed that a tomographic setup of six cameras is in principle suited to measure the main statistical characteristics of the puff dispersion in the ABL. However, the data set was limited by several points: 1) Artefacts from clouds in the image

are falsely interpreted as $SO_2$ making an automatic $SO_2$ retrieval difficult. For the data amounts necessary for a meaningful statistical analysis of puff releases, the data set should contain cloud free data to enable automatic retrieval. 2) Some propagation directions might get systematically discarded during the data processing. This would lead to an underestimation of the puff meandering. 3) The release of the $SO_2$ puffs is only nearly instantaneous, leading to elongated puffs. This puts an uncertainty on the relative dispersion estimate, in particular for the along-wind coordinate.

It is desirable to determine a value for the Richardson-Obukhov constant and the higher moments of the concentration distribution in order to constrain atmospheric turbulence models. A robust estimate for the Richardson-Obukhov constant of relative dispersion and Lagrangian integral time scales could be obtained from a larger data set of longer tracked single puffs. Such a data set is planned to be produced during follow-up COMTESSA field campaigns. The same concept as for the first campaign should be used but on a larger scale i.e. releasing larger amounts of $SO_2$. Higher amounts of $SO_2$ will increase

the images' signal-to-noise ratio and facilitate observations at larger distances to the tower. This increases consequently the cameras' field-of-view enabling puff observations over longer distances and times.

Further, several conclusions regarding the camera placement could be drawn from the first campaign: 1) Cameras should not observe the puffs frontal as it is impossible to separate overlapping puffs in the analysis. Alternatively the time between two releases has to be sufficiently long to allow a clear puff separation. 2) If possible, release experiments should only be performed

on cloud-free days or at least the cameras have to be positioned such that the clouds do not appear on the projected trajectories of the puffs. 3) Further, it should be possible to observe all propagation directions of the puffs to avoid biases in the meandering towards a certain direction. The used half-circle offers a good solution.

In the case of a cloud free data set, the presented method can be applied fully automatically. Hence, providing a larger and cloud free data set, opens the door for statistical analysis of puff dispersion. Further under cloud free conditions, the underlying

imagery can be used to conduct a complete tomographic reconstruction of $SO_2$ concentration, which will be invaluable for constraining models of atmospheric boundary-layer dispersion.

*Code and data availability.* The raw measurement data and the python code used for data analysis is available from the authors upon request. The code is based on the pyplis toolbox (Gliß et al., 2017)





## Appendix A: Details on the artificial release experiment

### A1  Reconstruction of the setup

Precise knowledge of the experimental setup is necessary for the reconstruction of 3D trajectories. During the field campaign, the distances of the cameras to the release tower and the angle towards north were measured using a theodolite. Comparing the pixel coordinate of the top of the release tower in the camera image with the tower's position, the three angles defining the camera pose (azimuth, elevation and tilt) were extracted. The results are shown in table A1.

### A2  Camera temporal synchronisation

As no GPS time information was yet available during the experiment, the image time series of the six $SO_2$ cameras had to be synchronised manually after the experiment. For this end, the release time of 18 subsequent puff releases between 10:29 and 10:31 were detected for every camera. Due to the distinct movement of the puffs within the turbulent flow, the puffs could be clearly correlated in the images of all cameras. The relative temporal offset $\Delta t_i$ between camera $i$ to camera 1 was then calculated from the time difference of the first frame, on which a puff was visible.

$$\Delta t_i = t_{i,start} - t_{1,start} \tag{A1}$$

The temporal offset was averaged over 18 observed puffs between 10:29 and 10:31 and is given relative to camera 1 in Table A2. The accuracy of the temporal offset is limited by the discrete sampling frequency which in turns is constrained by the exposure and readout time.

### A3  Data processing of the eddy covariance measurements

Five minutes of meteorological measurement between 10:27 and 10.32 UTC have been used to obtain the parameters reported in Table 2. Before the actual post processing, the collected data was treated by the LICOR EddyPro software system for despiking (e.g. Vickers and Mahrt, 1997; Mauder, 2013) and for applying the triple rotation correction (Wilczak et al., 2001) that nullify the average vertical and across-wind components, and the $\overline{v'w'}$ Reynolds stress component. This means that the coordinate system is aligned with the measured mean wind direction. See also Burba (2013) for a description of the corrections applied in EddyPro.

The values for the mean wind $\overline{u}$ and the three turbulent fluxes $\sigma_u^2, \sigma_v^2, \sigma_w^2$ are reported at 8.7 m close to the source location. The friction velocity $u_*$ was estimated by using the Reynolds stress component at two meters as $u_*^2 = |\overline{u'w'}|$. The Obukhov length L is defined as

$$L = -\frac{u_*^3 \overline{\theta_v}}{\kappa g \overline{w'\theta_v'}}, \tag{A2}$$

where $\theta_v$ is the virtual potential temperature, $\kappa \approx 0.4$ is the von Kármán constant, $g$ is the gravitational acceleration, and $\overline{w'\theta_v'}$ is the vertical turbulent flux of virtual potential temperature. We used the sonic temperature as an approximation of





virtual temperature as discussed in e.g. Kaimal and Finnigan (1994). As a consistency check, the flux Richardson number was calculated at $z$=5.4 m using

$$R_f = \frac{\overline{w'\theta'_v}\frac{g}{\theta_v}}{\overline{u'w'}\frac{\delta\overline{u}}{\delta z}}. \tag{A3}$$

In convective conditions, the flux Richardson number has a similar value to z/L (here, -0.868) (e.g. Stull, 1988) and our
measurements ($R_f$=-0.988) are in good agreement. The mean dissipation of turbulent kinetic energy $\epsilon$ was obtained by fitting a Kolmogorov spectrum $C_k\epsilon^{2/3}\kappa^{-5/3}$ to the inertial range of the measured spectrum for the along-wind component of velocity using the method discussed in detail by Stull (1988). The value of the Kolmogorov constant $C_k$=0.49 was taken according to measurements and theory of homogeneous isotropic turbulence (e.g Stull, 1988; Pope, 2000). The Eulerian integral time scale of the vertical velocity component $T_{E,w}$ was obtained by fitting an exponential decay to the autocorrelation function for
the measured five minute time series. The Lagrangian integral scale $T_{L,w}$ was estimated from the Eulerian one by using the empirical fixed ratio $\beta = \frac{T_{L,w}}{T_{E,w}}$ proposed by Hanna (1981) and Pasquill and Smith (1983), see Eq. (2) of the main manuscript.

## Appendix B: Details on image processing methods

### B1    Comtessa SO$_2$ slant column density retrieval

The raw intensity images have to go through several retrieval steps to get the final product, the SO$_2$ slant column densities. For
a detailed, general description see e.g. Kantzas et al. (2010) or Lübcke et al. (2013). In the following all images are corrected for the dark signal, which was recorded daily after the release experiments.

     Sky masks are defined for every camera based on local intensity thresholds. The sky masks separate the images in two regions according to whether the intensity contains a reflected component or only backscattered sunlight. Sunlight can be reflected from the ground, topography in the background, and structures such as the release tower and antennas. This reflected
region is completely ignored in the further analysis.

     The optical density images of the SO$_2$ puffs are calculated according to Eq. 16 from a SO$_2$-containing and SO$_2$-free background image. The SO$_2$-free background image is selected from the time series of puff releases. Typically, this image is cloud-free and can be scaled to the base intensity of an individual SO$_2$-containing image recorded at a later time (e.g. Gliß et al., 2017). However, due to the partly strong cloud cover, a background image containing the exact cloud structures but
no SO$_2$ is necessary for the analysis. Such an image cannot be scaled to the changing base intensity with time and is thus constrained to a short analysis period of few tens of seconds for quantitative analysis. Therefore, a "patchwork" image from the same time series during the puff release between 10:30:00 and 10:30:10 was selected for every camera. If a puff was present in this image, the respective image area was cut and replaced by the same area of an image several seconds later without the puff present in this area. The calibration from optical densities to SCDs is performed using the built-in DOAS spectrometer.





## B2    Algorithm description: Tracking of individual puffs in image plane

Figure A1 depicts the tracking algorithm schematically. The algorithm is based on three copies of the original image (see Fig. A2): (1) the original high-resolution image, (2) an image which was blurred with a 2D Gaussian function (mean: 1, sigma: 5) and (3) a low-resolution image which was sub-sampled to ($87 \times 65$ pixel) using Laplacian pyramids. The images are

increasingly noise-reduced and have consequently lower detection limits for $SO_2$. The average standard deviations for the three image types are (1) 2.4e16 molec cm$^{-2}$, (2) 1.75e16 molec cm$^{-2}$, and (3) 5.0e15 molec cm$^{-2}$.

The puffs are tracked iteratively from the release point. Therefore the image coordinates of the release point and the start image of the individual puffs have to be provided manually. The tracking will start from this image. After every successful detection of the ROI, the next image will be loaded. First the ROI is detected within the blurred image around the last-known

position of the puff. That is the release point for the first image, and the CM of the previous image for all other images. A $50 \times 50$ ROI is set around this point. Then the ROI is increased incrementally by single image rows and columns. New pixel rows or columns are added to the ROI if they contain at least 5% pixel above a threshold of 3.5e16 molec cm$^{-2}$. The threshold is chosen as the double of the standard deviation to suppress noise and cloud artefacts effectively. The ROI contains the central part of the puffs but not necessarily separated fractions and weak tails. Weak tails and separated fractions can be detected within the low-

resolution image which suppresses noise 4-times more compared to the blurred image. The image is separated into connected regions containing a significant signal. A pixel is considered to contain a signal if 25% of the pixels in a 5x5 neighbourhood are above a threshold. This methods detects the $SO_2$ puffs and clouds alike, thus a separate selection is necessary to identify the puffs. The detected ROIs are rescaled to the original resolution and compared to the previously detected ROI from the blurred image. If the previously found ROI immerses completely in a new ROI, it will be replaced by the larger ROI. In this way, the

full area of puffs including tails close to the detection limit and separated $SO_2$ patches are included. When the final ROI of a puff is determined, the total signal, CM and spread of the puff are calculated within this ROI based on the original image.

For the next image, the CM of the previous image is used as a starting point for the ROI which is determined equivalently. The procedure is repeated until an invalid ROI is detected. This is the case when the puff touches the image borders or moving in front of non-sky areas such as the ground or vegetation and topography on the horizon. In these cases, the ROI would no

longer contain the complete puff. Further, the tracking stops when it is likely that cloud artefacts are tracked instead of the puff. This can be indicated by jumps in the CM or a sudden increase or decrease of the ROI.

## B3    Sensitivity of trajectory retrieval to single camera

The 3D CM trajectories are calculated by triangulation based on the individual 2D CM trajectories of the six cameras. While using a least-square method including all six cameras reduces effects from uncertain camera position and pose and clouds, data

from only two cameras would be in principle sufficient for reconstructing the 3D trajectory. To determine the sensitivity to possibly inaccurate data obtained from certain cameras, we repeated the trajectory retrievals excluding systematically information from one camera (Fig. A3). The retrieved 3D trajectories show no particular sensitivity to a single camera view, suggesting that none of the cameras adds crucial or false information to the reconstruction. Excluding the data from the cameras containing the





most pronounced cloud cover (3,5,6) does not shift the retrieved trajectories outside the 1-$\sigma$-range of the trajectory including all cameras. Hence, we argue that information from such cameras can be used for the trajectory reconstruction even if they fail to fully detect and separate the puff from cloud artefacts.

**Appendix C: Videos of puff releases**

5    The online supplement contains videos of the six puff releases recorded with the six cameras. The videos are available at the online respository Zenodo: Dinger, Anna Solvejg. (2018). Videos of artificially released SO2 puffs recorded simultaneously with six UV SO2 cameras. Zenodo. http://doi.org/10.5281/zenodo.1299638

The detected ROI and CM are indicated on every image frame. The images were noise-reduced (Gaussian filter with $\sigma$=5) to increase the visibility of the puffs for the human eye. Note that the influence of cloud cover becomes more evident as the
10    time difference between background image and image frame increases. Further, the times of the background images can be seen in the video: The image background noise cancels to zero for this time according to Eq. (16). In some videos, additional absorption from small insects flying through the cameras' field of view are visible in the form of straight lines.

*Author contributions.* ASD, AS and MC wrote the manuscript. KS, MC, AK, AS contributed with discussion to the manuscript. ASD analysed the camera data and developed the methodology. MC, HA, and SYP analysed the Eddy-Covariance measurements towards turbulence.
15    AS, MC and KS designed the Comtessa experiment. ASD, KS, MC, HA, SYP, NS, JW and AS contributed to the field experiment. The SO$_2$ cameras have been designed by CB, KS and developed by KS, ASD and CB

*Competing interests.* The authors declare that no competing interests are present.

*Acknowledgements.* The COMTESSA project has received funding from the European Research Council (ERC) under the European Union's Horizon 2020 research and innovation programme under grant agreement No 670462.





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



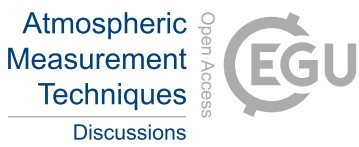
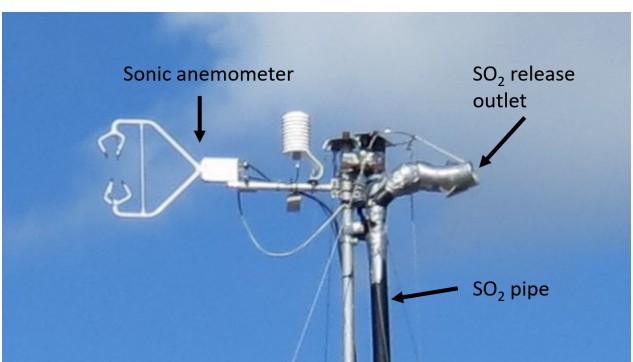

**Figure 1.** Top of the release tower.





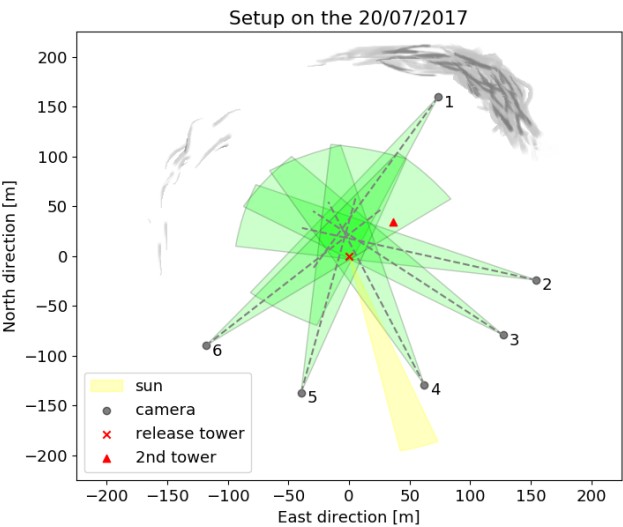

**Figure 2.** Map of the experimental setup. The cameras' FOV are indicated in green. The sun position (yellow) and the cloud cover (gray) as observed at 10:30 UTC are sketched on the map. Coordinates are given relative to the release location.

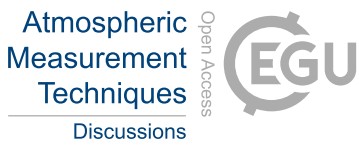



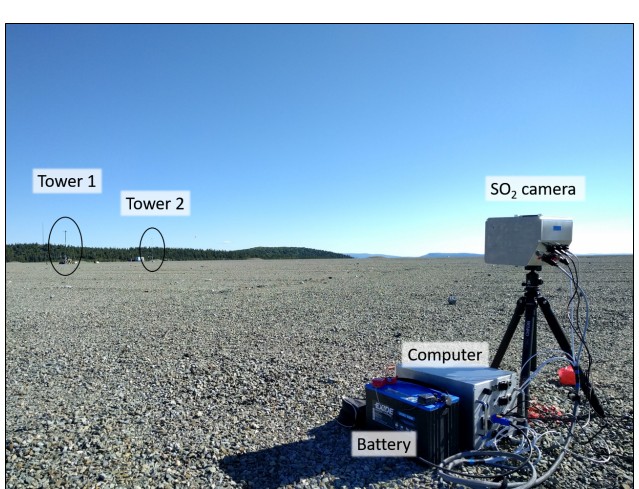

**Figure 3.** SO$_2$ camera and PC, here camera 5. In the background, the release and measurement towers are visible.





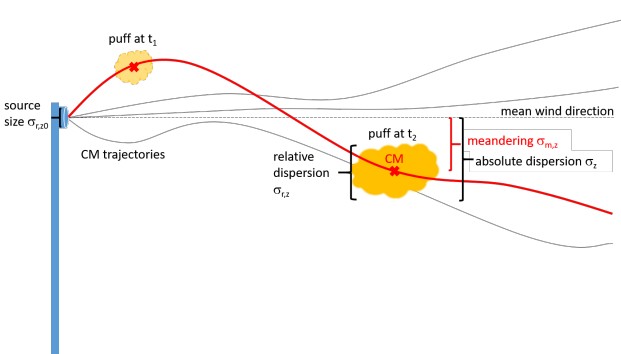

**Figure 4.** Sketch of a puff release. The centre of mass trajectory of a single puff (red) meanders around the mean trajectory of a puff ensemble, while the puff additionally spreads around it centre of mass. Consequently, the absolute dispersion can be separated into the meandering of the centre of mass trajectories and the variance of the puffs' concentration relative to their centre of mass. Both are obtained using data from a large number of realizations of single puff releases.




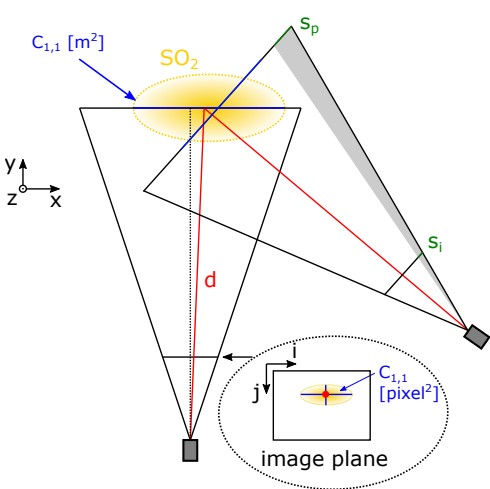

**Figure 5.** Sketch of the field of view of two cameras from above. The three-dimensional $SO_2$ puff (yellow) in the world coordinate system (x,y,z) is projected to the two-dimensional image plane (i,j). The centre of mass in the image plane corresponds to a solid angle in the world coordinate system (red). The apparent size of a pixel scales with the distance to the object plane (grey area).





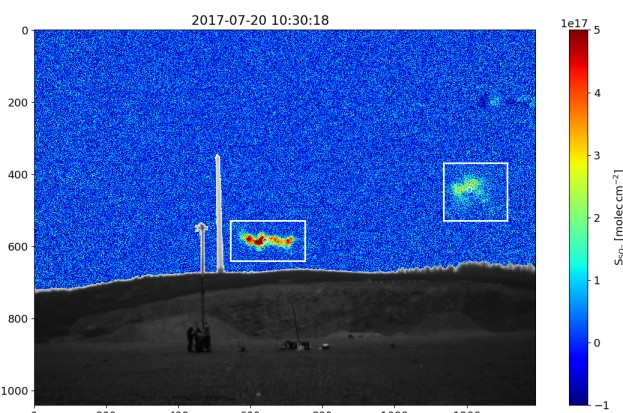

**Figure 6.** Example of a SO$_2$ CIC image from a SO$_2$ camera, here camera 4. The image contains two SO$_2$ puffs marked by the detected ROI (white rectangle). Artefacts produced by a cloud are visible in the upper right corner.





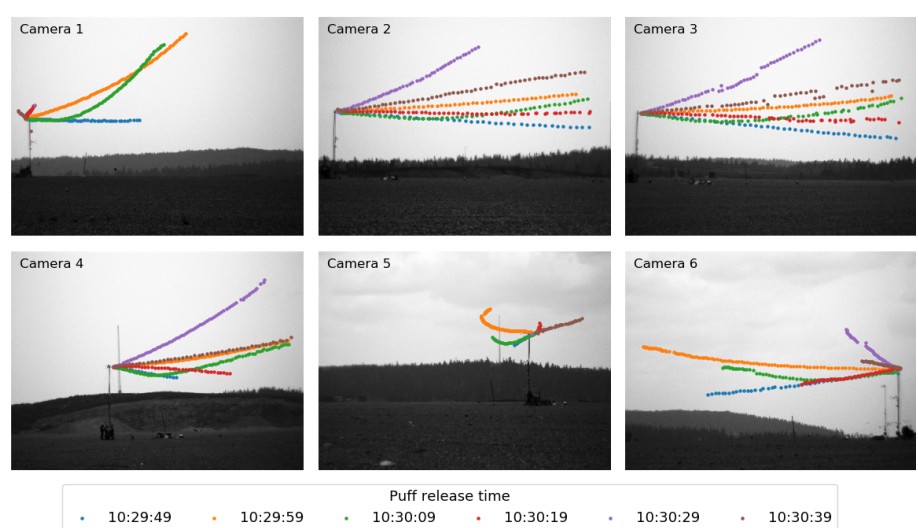

**Figure 7.** Centre of mass coordinates of six subsequent puffs projected to the image planes of the six SO$_2$ cameras. For cameras 4 and 5, a meteorological tower is visible in the image background. This tower is located a few hundred meters northwest of the release tower.





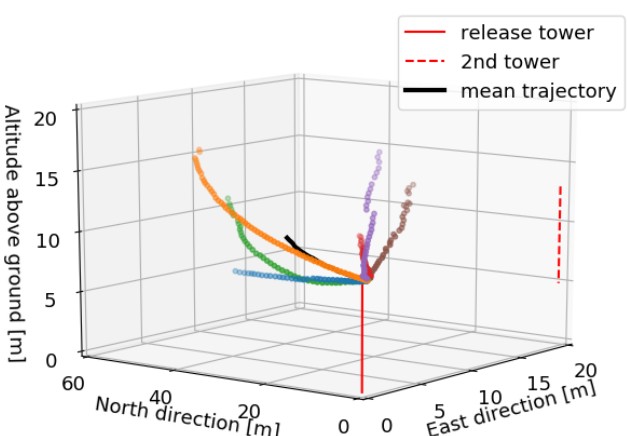

**Figure 8.** CM trajectories of the six observed puffs and the ensemble average.





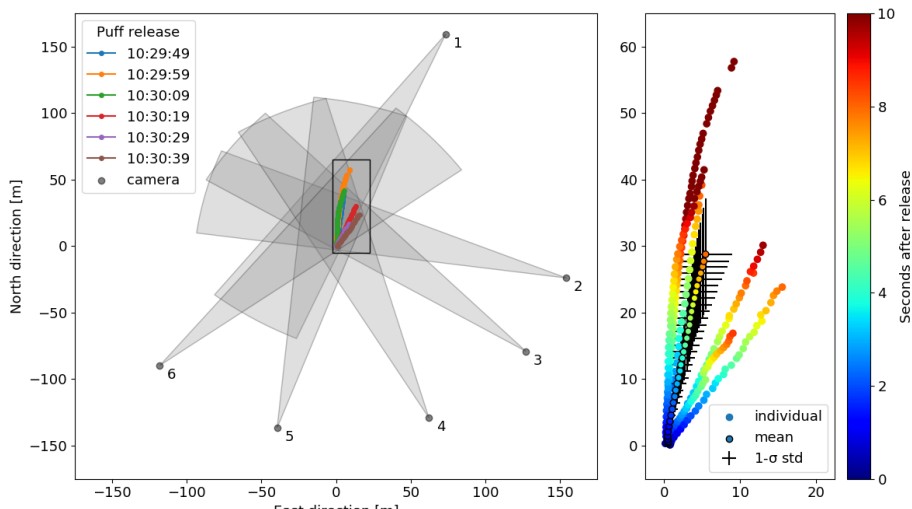

**Figure 9.** Horizontal projection of the CM trajectories of the six puffs observed with six $SO_2$ cameras. The left panel shows an overview of the camera positions relative to the reconstructed trajectories. The right panels shows a blow-up of the rectangular area marked in the left panel. The colour code represents the travel time since release. The mean trajectory and its standard deviation are displayed with black pluses.





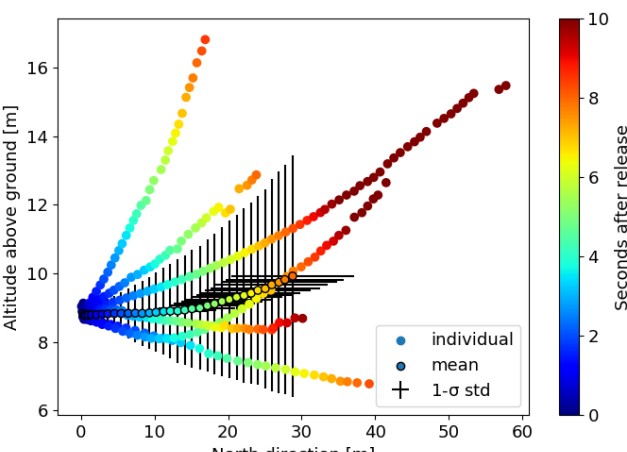

**Figure 10.** Vertical projections of the CM trajectories to the altitude-north plane. The colour code represents the travel time since release. The mean trajectory and its standard deviation are displayed with black pluses. Note that the x-axis scales 6× larger than the y-axis.





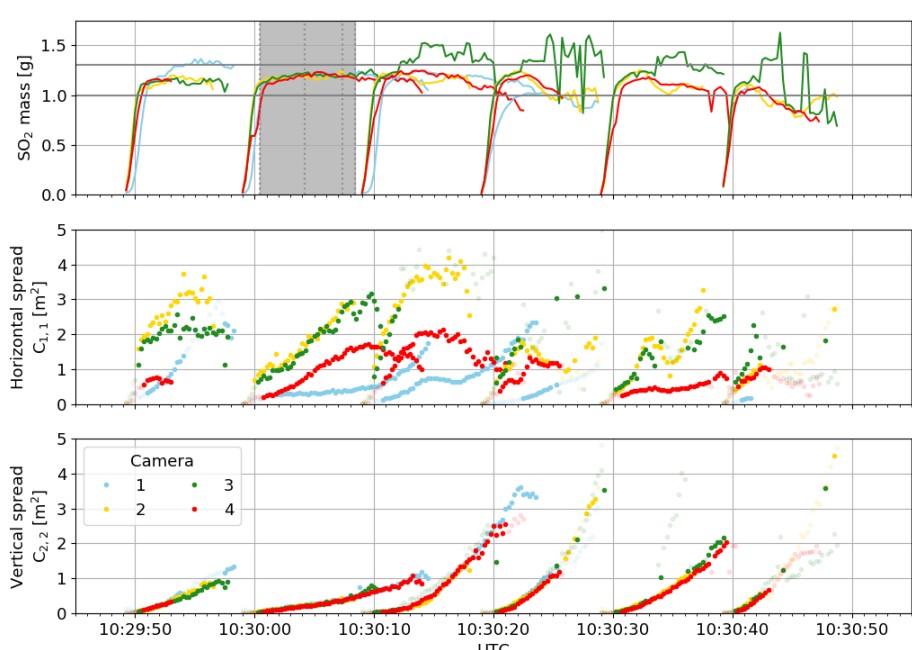

**Figure 11.** Total mass, horizontal spread and vertical spread (lower panel) of six subsequent puffs. Only data from cameras 1-4 are shown due to significant cloud signals in camera 5 and 6. The background images, and thus cloud cover, were reconstructed from the shaded time period. The shaded data points are discarded because their corresponding mass lies outside the expected range (1.0-1.3 g).





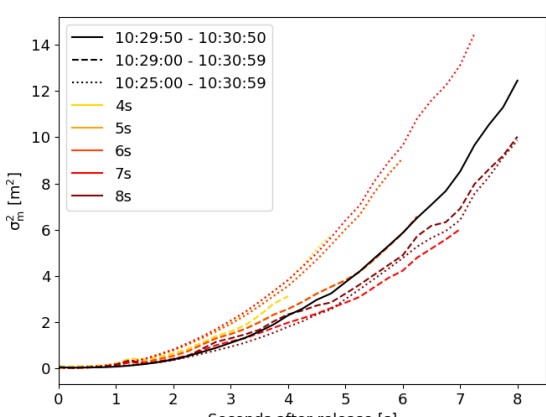

**Figure 12.** Meandering in the vertical. The black curve shows the ensemble average over the six puffs. The meandering is sensitive to the chosen ensemble. The coloured dashed and dotted curves show the meandering calculated for different numbers of puffs, selected by varying the time interval (line style) and the minimum trajectory lengths (colour).





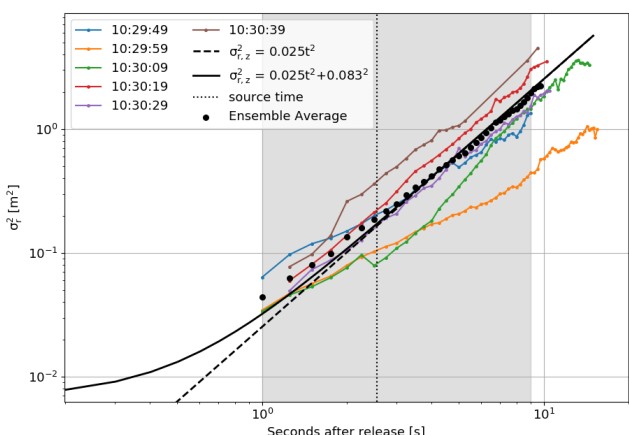

**Figure 13.** Relative dispersion in the vertical on a log-log-scale. The coloured curves show the dispersion of individual puffs and the black points show the ensemble average over these individual puffs. The source size was estimated by a linear fit to the ensemble average (dashed black line). The resulting source size was used to calculate the predicted curve by Eq. (4) (solid line) and estimate the source time (dotted black line).



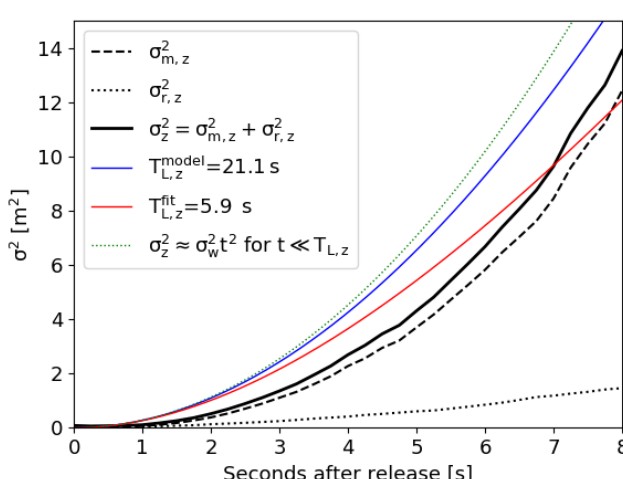

**Figure 14.** Absolute dispersion in the vertical (solid black line). Relative dispersion and meandering are shown with dotted and dashed black lines, respectively. Two parametrisations of Taylor's theorem are plotted: 1) modelled from the sonic anemometer data (blue) and fitted to the measured absolute dispersion (red). For dispersion times much smaller than the Lagrangian time scale, the absolute dispersion can be approximated by a $t^2$-dependency (green).



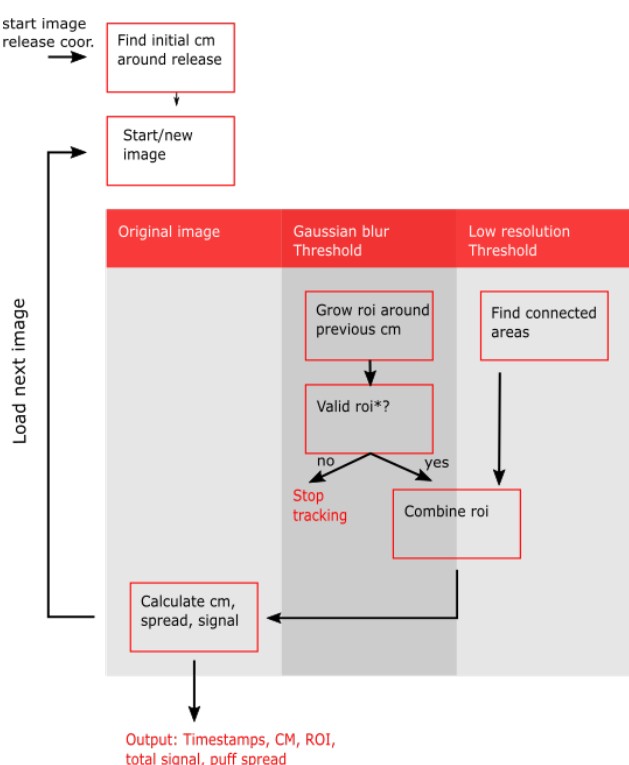

**Figure A1.** Flow diagram of the tracking algorithm. The puffs are detected iteratively based on the previous detection and two noise-reduced versions of the original image. The conditions for a valid ROI can be found in the text.





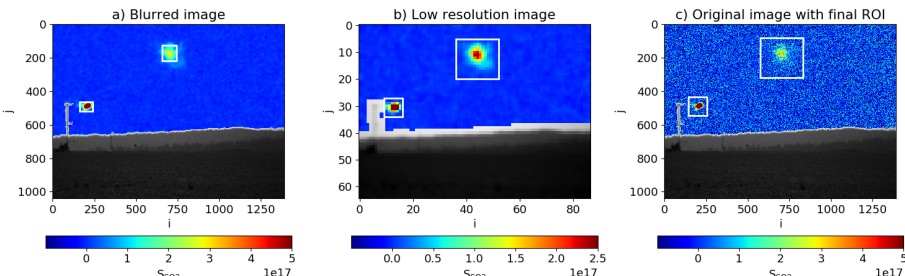

**Figure A2.** Puff detection based on noise-reduced images, here for camera 1 at 10:30:12. The ROI is detected in a blurred image based on the position of the CM in the previous image (a). A low resolution image is used to detect connected areas above a threshold (b). The combination of both detections gives the resulting ROI, which is used to calculate the CM, total signal and spread in the original image (c).





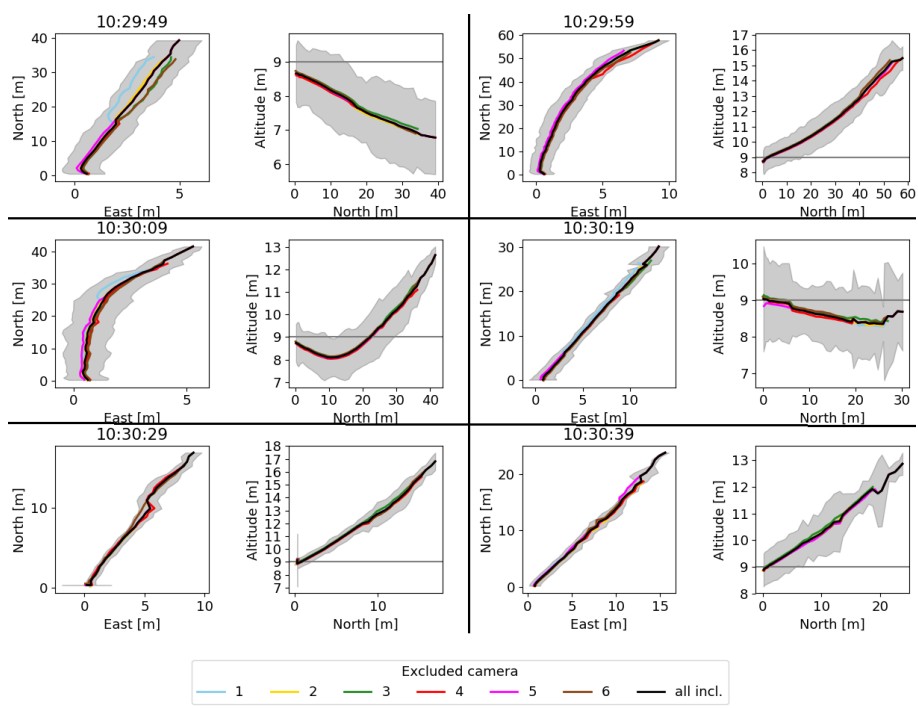

**Figure A3.** Sensitivity of the reconstructed trajectories to the removal of data from a single camera. The trajectory colour indicates which camera was removed from the calculation, the black trajectory is based on data from all cameras. The time indicates the release time of the puff. In the altitude-north plots, the horizontal line represents the release altitude.



**Table 1.** Summary of SO$_2$ camera properties.

| Property | | |
|---|---|---|
| pixel number | $N_i \times N_j$ | $1392 \times 1040$ |
| pixel size | $s_i \times s_j$ | $4.65\mu$m $\times$ $4.65\mu$m |
| focal length | $f$ | 25.06 mm ($\lambda$=266 nm) |
| field of view | | $14.7° \times 11.1°$ |
| filter wavelength | $\lambda$ | 310 nm and 330 nm |





**Table 2.** Measured turbulence parameters at 10:27 - 10:32 UTC

| Parameter | Symbol | Value |
|---|---|---|
| *direct measurements* | | |
| mean wind velocity | $\overline{u}$ | 5.22 m s$^{-1}$ |
| fluctuations, along-wind | $\sigma_u^2 = \overline{u'^2}$ | 2.29 m$^2$ s$^{-2}$ |
| fluctuations, across-wind | $\sigma_v^2 = \overline{v'^2}$ | 0.861 m$^2$ s$^{-2}$ |
| fluctuations, vertical | $\sigma_w^2 = \overline{w'^2}$ | 0.283 m$^2$ s$^{-2}$ |
| turbulence intensity | i | 0.102 |
| Obukhov length | $L$ | -6.22 m |
| flux Richardson number | $R_f$ | -0.988 |
| friction velocity | u$_*$ | 0.249 m s$^{-1}$ |
| *fit to energy spectra* | | |
| energy dissipation | $\epsilon$ | 0.015 m$^2$ s$^{-2}$ |
| Eulerian integral time, vertical | $T_{E,w}$ | 3.07 s |
| Lagrangian integral time, vertical | $T_{L,w}$ | 21.1 s |
| Ratio of integral times | $\beta$ | 6.87 |



**Table A1.** Camera pose and position

| Camera | distance (x,y,z) [m, m, m] | azimuth [°] | elevation [°] | tilt [°] |
|---|---|---|---|---|
| 1 | (73.5, 159.6, 0.8) | 210.9 | 2.5 | 0.0 |
| 2 | (154.3, -24.0, 0.8) | 285.1 | 2.6 | 0.1 |
| 3 | (127.1, -79.1, 0.8) | 308.4 | 2.8 | 0.6 |
| 4 | (62.2, -129.2, 0.8) | 336.6 | 3.9 | -1.4 |
| 5 | (-39.5, -136.8, 0.8) | 13.0 | 2.3 | -0.1 |
| 6 | (-118.1, -89.8, 0.8) | 46.6 | 3.9 | -1.0 |



**Table A2.** Relative differences in recorded (system) time stamps and exposure times (at 10:30)

| Camera | $\Delta t_i$ [s] | $\Delta t_{exp}$ [s] |
|---|---|---|
| 1 | - | 0.16 |
| 2 | $0.15 \pm 0.10$ | 0.16 |
| 3 | $6.23 \pm 0.08$ | 0.17 |
| 4 | $6.28 \pm 0.08$ | 0.16 |
| 5 | $1.04 \pm 0.08$ | 0.20 |
| 6 | $0.61 \pm 0.10$ | 0.17 |