# Peer review of "Observation of turbulent dispersion of artificially released SO2 puffs with UV cameras"

_Atmospheric Measurement Techniques, 2018_

## Referee Comment (RC1) · J-F. Smekens (Referee) · 13 Aug 2018

This manuscript describes the results of a large scale experiment of to sample the three-dimensional (3D) concentration distribution of an atmospheric tracer (sulfur dioxide – SO2) in the atmospheric boundary layer at high spatial and temporal resolution, using a network of UV cameras. UV cameras are increasingly used in volcanology research to quantify SO2 emissions from a variety of eruptions. This application however, represents an innovative use for the instrument, and further demonstrates its advantages for atmospheric research in general. The uniqueness of the experiment makes their findings extremely valuable to the community, and the authors detail those findings

with very clear phrasing and comprehensive figures. I strongly recommend the publication of this manuscript and have only a few general comments and recommendations that could improve the general discussion.

General comments

Continuous release experiment. The authors mention experiments with continuous release of SO2 (both in the text Px,Lxx and in the abstract P1,L7). Yet no results are shown or even discussed from that set of experiments. Given the added value that such a dataset would represent, especially to members of the volcanology community, I would suggest the authors either include some results (even if they are not entirely conclusive) in their manuscript, or explicitly state why they will not be discussed.

On the use of tomography. The authors correctly state that to this day, no successful tomography has been reported with UV camera imagery. The presented study, though very compelling and entirely justified, still does not present tomography results. The imagery is used to project trajectories for the center of mass of each puff, and calculate spread and dispersion factors. The full inverse problem yielding a 3-D concentration map of a puff remains unsolved. Perhaps a clarification to this point could be added in the discussion?

Specific comments

P3, L15 – Just a small note. Although clear sky conditions will provide a higher UV signal, this signal remains non-uniform. Excellent acquisition conditions can be obtained on cloudy days if the cloud cover is uniform at a sufficiently high ceiling. Problems arise when the cloud cover is either very low or non-uniform (i.e. scattered clouds).

P9, L27 – What specific techniques were used for noise reduction of the images? This could be added to the Appendix.
* * *

---

## Referee Comment (RC2) · Anonymous Referee #2 · 16 Aug 2018

The authors present an interesting new measurement technique with great potential of improving knowledge on turbulent dispersion. That being said, the data set is suboptimal to give a full demonstration of the power of the method (see my second comment below), and I wonder if this paper should be published at this stage, and not after obtaining better data. The analysis feels rather quick and rough, and I would prefer a more in-depth paper, because it is nearly impossible to judge the full quality of the method here. Furthermore, some of the turbulence theory need to be motivated in more detail.

Can you also specify what is the specific contribution of all authors to the actual results in the paper? I am not sure after reading the author contributions whether all of those

warrant co-authorship of the paper.

Specific comments:

P2, L7: What do you mean with the PDF has large fluctuations? Do you mean that the flow has large fluctuations, resulting in a wide PDF?

P2, L10: Your statement on direct numerical simulation is not correct. In order to produce meaningful DNS, it is not necessary to reach similar Reynolds numbers as in the atmosphere. Many of the statistics of the flow converge with Reynolds numbers far below those in the atmosphere, as can for instance be seen in van Heerwaarden & Mellado (2016, JAS), who show converging statistics in DNS a convective boundary layer. I consider the authors to have a look at the seminal paper of Moin and Mahesh (1998) and the appropriate interpretation of DNS as a research tool.

P3, L23-30: I get a little uncomfortable from this paragraph. Not all equipment was operational yet, your dataset contains a too small number of puffs for meaningful statistics, and later in the paper (P4 L8) you also refer to sub-optimal weather conditions. Why did you choose to publish this paper now, and not after you have obtained better data? As the method is so promising, why wouldn't you wait?

P6, eq 4: This theory only applies if the puffs have length scales far less than the production scales of the flow, as well as the dissipation scale. Can you give those numbers for the flow your experiment is embedded in?

P6, eq 5: Is the turbulence isotropic at the scales you are looking? Are the variances of the three flow components on the time scale of dispersion the same?

P12, L29-31: What is the exact interpretation of the t2 and t3 regime. You could make the link with eqs 4-6, and explain the physical meaning better.

Figures: Please make the figures such that they all have a consistent, and readable font size.

---

## Author Comment (AC1) · 21 Sep 2018

**Author comment to Referee comment #1 "Excellent application of a novel method" on «Characterising vertical turbulent dispersion by observing artificially released SO2 puffs with UV cameras"**

We thank Jean-François Smekens for his thorough review and helpful suggestions. Please find our answers in blue underneath each point raised by the reviewers. Resulting manuscript changes are stated below our answers with changed passages marked in red.

This manuscript describes the results of a large scale experiment of to sample the three-dimensional (3D) concentration distribution of an atmospheric tracer (sulfur dioxide – $SO_2$) in the atmospheric boundary layer at high spatial and temporal resolution, using a network of UV cameras. UV cameras are increasingly used in volcanology research to quantify SO2 emissions from a variety of eruptions. This application however, represents an innovative use for the instrument, and further demonstrates its advantages for atmospheric research in general. The uniqueness of the experiment makes their findings extremely valuable to the community, and the authors detail those findings with very clear phrasing and comprehensive figures. I strongly recommend the publication of this manuscript and have only a few general comments and recommendations that could improve the general discussion.

General comments

Continuous release experiment. The authors mention experiments with continuous release of SO2 (both in the text Px,Lxx and in the abstract P1,L7). Yet no results are shown or even discussed from that set of experiments. Given the added value that such a dataset would represent, especially to members of the volcanology community, I would suggest the authors either include some results (even if they are not entirely conclusive) in their manuscript, or explicitly state why they will not be discussed.

> The data from the continuous releases are not yet fully analysed, partly because the plume meteorological conditions during these experiments were unfavourable or, on 20 July, the plume measurements were carried out early in the morning, when UV light levels were still low. However, in principle, the results that can be obtained from the plume data set are of relevance for both the volcanology community (e.g. validation of $SO_2$ flux retrieval, radiative transport effects) and the turbulence community (e.g. plume dispersion studies). We have conducted a new campaign this summer under more favourable conditions and will also analyse plume experiments. However, this will take time, and is out of scope of the present paper. We have added an explanation that plume experiments are not studied in the present paper.

> P4 L20: "In this paper, however, only analyses of the puff experiments will be presented."

On the use of tomography. The authors correctly state that to this day, no successful tomography has been reported with UV camera imagery. The presented study, though very compelling and entirely justified, still does not present tomography results. The imagery is used to project trajectories for the center of mass of each puff, and calculate spread and dispersion factors. The full inverse problem yielding a 3-D concentration map of a puff remains unsolved. Perhaps a clarification to this point could be added in the discussion?

> We agree with the referee and we will clarify this point in the manuscript. Although a tomographic setup which in principle allows for a full tomography was used, no full tomography results are actually presented in this study. First tests of a full tomographic reconstruction have shown that during this campaign the camera positions and especially the time synchronisation of the cameras, were not accurate enough for a 3D reconstruction. These issues have been solved in the meantime, so we expect full tomography studies to be possible in the future.

> P3 L30: "However, note that a fully resolved tomographic reconstruction is not necessary for this retrieval and is not presented in this paper."

Specific comments

P3, L15 – Just a small note. Although clear sky conditions will provide a higher UV signal, this signal remains non-uniform. Excellent acquisition conditions can be obtained on cloudy days if the cloud cover is uniform at a sufficiently high ceiling. Problems arise when the cloud cover is either very low or non-uniform (i.e. scattered clouds).

> We agree with the referee. Unfortunately, during our experiments the clouds were always either inhomogeneous or quite low, or both.

> P3 L15: "Non-uniform cloud cover in the image background can cause inhomogeneous illumination of the sky, which complicates the $SO_2$ column concentration retrieval of $SO_2$ camera images. "

P9, L27 – What specific techniques were used for noise reduction of the images? This could be added to the Appendix.

> The specific techniques are described in Appendix B2. The images are smoothed with a Gaussian filter, and separately subsampled in order to reduce the noise.
The Gaussian filter was performed using the function cv.GaussianBlur() from the OpenCV library. The kernel size was 5 and the sigma was 1.

The subsampling is internally performed using Laplacian image pyramids which allow to switch between different compression levels of the image. Explicitly, each compression step consist of a Gaussian Filter (5x5 kernel size) and  subsampling by rejecting even rows and columns. Each such step reduces the pixel number by a factor of 4. To reach a resolution of 87x65 pixel, four compression steps were subsequently performed on the images by applying the OpenCV function cv.pyrDown() four times.

> Appendix B2: "The algorithm is based on three copies of the original image (see Fig. A2): (1) the original high-resolution image, (2) an image which was blurred with a 2D Gaussian function (mean: 1, sigma: 5) and (3) a low-resolution image which was sub-sampled to (87x65 pixel) using image pyramids. The images are increasingly noise-reduced and have consequently lower detection limits for $SO_2$ The average standard deviations for the three image types are (1) 2.4e16 molec cm$^{-2}$, (2) 1.75e16 molec cm$^{-2}$, and (3) 5.0e15 molec cm$^{-2}$"

---

## Author Comment (AC2) · 21 Sep 2018

**Author comment to referee comment #2 "Interesting and promising technique, too preliminary results" to comments on «Characterising vertical turbulent dispersion by observing artificially released SO$_2$ puffs with UV cameras"**

We thank referee #2 for their thorough review and helpful suggestions. Please find our answers in blue underneath each point raised by the reviewer. Resulting manuscript changes are stated below our answers with changed passages marked in red.

The authors present an interesting new measurement technique with great potential of improving knowledge on turbulent dispersion. That being said, the data set is suboptimal to give a full demonstration of the power of the method (see my second comment below), and I wonder if this paper should be published at this stage, and not after obtaining better data. The analysis feels rather quick and rough, and I would prefer a more in-depth paper, because it is nearly impossible to judge the full quality of the method here. Furthermore, some of the turbulence theory need to be motivated in more detail.

> We agree with the referee that our data set is suboptimal. Nevertheless, we believe that our study is worth publishing, for several reasons.

1) The study presents the measurement instrumentation, the experiment set-up and a proof-of-concept analysis of parts of the data set. All of that is novel and we believe that this is worthy of a publication, as confirmed by the other reviewer. The fact that the analysis of the turbulent dispersion is limited is also the reason why we have submitted this study to Atmos. Meas. Tech. and not to a journal with a scope encompassing the theory of atmospheric turbulence (e.g., Bound. Lay. Met.). However, we agree that the title and abstract might evoke too strong expectations in the reader towards turbulence results. We added a not in the abstract and conclusion to clarify its scope (see manuscript changes below). We have furthermore changed the title to "Observation of turbulent dispersion of artificially released SO$_2$ puffs with UV cameras" to more clearly emphasize the focus on the observations.

2) The SO$_2$ retrieval from the image data takes careful analysis and has not been done for discrete SO$_2$ puffs before. Also the simultaneous recording from different directions and the correction of non-uniform clouds in the background are current research topics. Such information in itself is interesting for researchers applying UV cameras, for example for volcano monitoring.

3) Due to the toxicity of sulfur dioxide (at high concentrations), release experiments have to planned far in advance and in cooperation with local authorities. Furthermore, the military compound that we are allowed to use, is only available for a few weeks in July when the military is not using the site. That means that we cannot easily repeat experiments or extend the

measurement period. In summer 2017, we have also been rather unlucky with the weather conditions during the experiment, which made the analysis of the data very difficult. At the same time, waiting with the publication until better data are obtained would have been unwise, since other researchers are following our work and want to be informed.

> Title: "Characterising vertical turbulent dispersion by observing artificially released SO₂ puffs with UV cameras" modified to "Observation of turbulent dispersion of artificially released SO₂ puffs with UV cameras"

> Abstract, P1 L14: "In this paper, we present a feasibility study demonstrating that the turbulent dispersion parameters can be retrieved from images of artificially released puffs, although the presented data set does not allow for an in-depth analysis of the obtained parameters. … In principle, the Richardson-Obukhov constant of relative dispersion in the inertial subrange could be also obtained, but the observation time was not sufficiently long in comparison to the source time scale to allow an observation of this dispersion range."

> Conclusions, P14 L7: "As a proof-of-concept, the absolute dispersion, as well as the relative dispersion and meandering of an ensemble of six puffs could be retrieved by performing a reconstruction of the 3D trajectories of the centre of mass positions of instantaneous puff releases."

Can you also specify what is the specific contribution of all authors to the actual results in the paper? I am not sure after reading the author contributions whether all of those warrant co-authorship of the paper.

> The campaign was a huge group effort and all authors have carried out significant tasks in the design, development, execution and/or analysis of the release experiment and the acquired data set.
We apologize, that we forgot to mention the contribution of Ignacio Pisso at the "author contribution" paragraph. He analysed the optimal positioning of the cameras before the experiment, as well as testing the tomography algorithm with the presented data.

> Autor contributions: "KS, MC, AK, AS and IP contributed with discussion to the manuscript. IP modelled the optimal setup of the UV cameras."

Specific comments:

P2, L7: What do you mean with the PDF has large fluctuations? Do you mean that the flow has large fluctuations, resulting in a wide PDF?

> We meant that the scalar field exhibits large fluctuations. We clarified the phrase.

> P2 L6: "A substance (a "passive scalar") injected into a turbulent atmospheric flow exhibits complex dynamical behaviour. Its distribution is stochastic, and the probability density function (PDF) of the scalar

concentration field exhibits the signature of large fluctuations, and can depart substantially from Gaussian behaviour (e.g. Shraiman and Siggia, 2000)."

P2, L10: Your statement on direct numerical simulation is not correct. In order to produce meaningful DNS, it is not necessary to reach similar Reynolds numbers as in the atmosphere. Many of the statistics of the flow converge with Reynolds numbers far below those in the atmosphere, as can for instance be seen in van Heerwaarden & Mellado (2016, JAS), who show converging statistics in DNS a convective boundary layer. I consider the authors to have a look at the seminal paper of Moin and Mahesh (1998) and the appropriate interpretation of DNS as a research tool.

> Our sentence (P2, L10) "The direct numerical simulation of turbulence (Orszag and Patterson, 1972) is not feasible at Reynolds numbers typical for the atmospheric boundary layer (ABL), and …" is actually true and correct, i.e. DNS is not feasible at Reynolds numbers typical of the ABL. However, we agree with the reviewer that DNS is useful also at lower Reynolds numbers as some turbulence statistics converge at relatively low Reynolds number.

The reviewer mentions the work of van Heerwaarden & Mellado (2016) that observe convergence of some Eulerian statistics up to second order at Taylor microscale Reynolds numbers between 150 and 180.

However, the use of DNS for studying scalar dispersion statistics and the Richardson regime in particular, is not straightforward and requires further clarification. The first thing to note is that Lagrangian dispersion statistics converge slowly with Reynolds number compared to Eulerian statistics. Yeung (2002) notes that the ratio of Lagrangian integral time scale to Kolmogorov time scale grows much slower with Reynolds number than any ratios of Eulerian length scales or timescales. Yeung et al. (2006) underline that if an inertial subrange exists in Lagrangian statistics it requires a higher Reynolds number to be clearly observed than in Eulerian statistics of similar order. Yeung et al. (2006) perform DNS simulations in the Taylor microscale Reynolds number range 40 to 650 and note that the Reynolds numbers in their data are still not sufficient to produce a fully unambiguous scaling range. Moreover, relative dispersion must ensure a sufficient range of scale between the source length and time scales and the integral scales that further increase the necessary extension of the inertial range and therefore the Reynolds number requirements. For example, the laboratory experiments of Ouellette et al. (2006) at Taylor microscale Reynolds number of 815 seems to have an insufficient Reynolds number, given the initial particle separation, to be able to observe the Richardson regime. Concluding, we think that our statement is correct but to avoid misinterpretation we extended it.

> P2 L9: "The direct numerical simulation of turbulence (DNS, e.g. Orszag and Patterson, 1972) is not feasible at Reynolds numbers typical for the atmospheric boundary layer (ABL). Although some Eulerian turbulence properties seem to converge also at relatively low Reynolds number (e.g. van

Heerwaarden & Mellado, 2016; Dimotakis, 2000), the Lagrangian dispersion statistics in general, and the relative dispersion in particular, require a high Reynolds number to converge and this poses challenges to both DNS and laboratory observations (e.g. Yeung, 2001; Yeung et al., 2006; Oluette et al., 2006). Other models used for tracer dispersion (e.g., Large Eddy Simulation or Lagrangian particle models) require parametrisations and/or validation based on atmospheric observations (e.g., Hanna 1984; Arya, 1999)."

P3, L23-30: I get a little uncomfortable from this paragraph. Not all equipment was operational yet, your dataset contains a too small number of puffs for meaningful statistics, and later in the paper (P4 L8) you also refer to sub-optimal weather conditions. Why did you choose to publish this paper now, and not after you have obtained better data? As the method is so promising, why wouldn't you wait?

> Please see the answer to the main comment above.

P6, eq 4: This theory only applies if the puffs have length scales far less than the production scales of the flow, as well as the dissipation scale. Can you give those numbers for the flow your experiment is embedded in?

> For Eq. (4) to be applicable (see also e.g. Monin and Yaglom, 1975, page 543, eq. 24.30') it is necessary that the initial source size is larger than the Kolmogorov length scale and smaller than the length scale of (local) energy containing eddies (i.e. source size is in the inertial subrange). Moreover, the time after release must be smaller than the characteristic time scale of the source, $t_S$, as already reported in the manuscript. Therefore, we are assuming here that the reviewer is interested in knowing if the initial source size is much smaller than local energetic eddies. We remind that the source size is estimated to be 0.083 m and the radius of the pipe is 0.0625 m. However, we are not completely sure what are the exact definitions of the length scales mentioned by the reviewer. We report here the definitions of the length scales that we are considering, according to our understanding:

- Local dissipation length scale based on vertical velocity: $l_\varepsilon = \sigma_w^3(z_s)/\varepsilon(z_s)$ = 10.2 m

- Production length scale: $l_P = kz_s$ = 3.48 m

Where $z_s$ (=8.7 m) is the source elevation. These are much larger than the initial puff size. We also estimated a local Eulerian integral time scale for the vertical velocity of about 3 s (see Table 2 in the manuscript). This corresponds to a length scale of about 15 m using Taylor's hypothesis. In any case, for Eq. (4) to be valid, it is most important that the turbulent spectrum shows an inertial range and that this range extends to scales much larger than the puff size (here between source size and 2 m). We examined the vertical velocity spectrum and found that the inertial range starts from a wavenumber corresponding to a length scale of about 9 m. Future experiments will use a

much higher source elevation, so that the length scales are larger to ensure a more  extended inertial subrange.

We changed the structure of the manuscript to stress beforehand that the presented Eq. (4)-(6) are only valid for the inertial subrange.

> P6 L11: "For an initial particle separation (puff size) in the inertial subrange of turbulence, i.e. larger than the Kolmogorov length scale and smaller than the length scale of (local) energy containing eddies, the particle separation will be first influenced by the source size and then become independent of the initial separation (e.g Monin and Yaglom, 1975; Franzese and Cassiani, 2007 Eq. A1-A6). Based on inertial range scaling arguments (e.g Monin and Yaglom, 1975), the characteristic time scale of the source is given by $t_s = (r_0^2/\varepsilon)^{1/3}$, where $\varepsilon$ is the mean dissipation of turbulent kinetic energy. The following Eq. (4)-(6) are valid for puff sizes in the inertial subrange of turbulence, which was observed in our experiment (see appendix A3 for details)".

P6, eq 5: Is the turbulence isotropic at the scales you are looking? Are the variances of the three flow components on the time scale of dispersion the same?

> At the scales that we are considering (see also answer to the point above), we observe a well-developed inertial subrange, $k^{-5/3}$, for all the three velocity components. The reviewer should note that we used isotropic arguments to estimate energy dissipation from the energy spectra and the estimates using different velocity components have maximum differences limited to about 30%. We think that this justify the use of local isotropic turbulence scaling arguments.

> Appendix A3, P16 L2: "The energy spectrum $E_i(k)$ of the $i^{th}$ velocity component, where $k$ is the wavenumber, is the Fourier transform of the autocorrelation function of that velocity component and was calculated according to e.g. Stull (1988, p.312) and using Taylor's hypothesis .…"

> Appendix A3, P16 L13: "The mean dissipation of turbulent kinetic energy $\varepsilon$ was obtained by fitting a Kolmogorov spectrum $E(k) = C_k \varepsilon^{2/3} k^{-5/3}$ to the inertial range of the measured spectrum for the along-wind component of velocity using the method discussed in detail by Stull (1988). The value of the Kolmogorov constant $C_k=0.49$ was taken according to measurements and theory of homogeneous isotropic turbulence (e.g. Stull, 1988; Pope 2000). We observe a well-developed inertial subrange starting at a length scale of about 9 m and the differences between estimates of $\varepsilon$ based on the three different velocity components are limited to about 30%."

P12, L29-31: What is the exact interpretation of the $t^2$ and $t^3$ regime. You could make the link with eqs 4-6, and explain the physical meaning better.

> Both regimes describe the relative dispersion for length scales in the inertial subrange of turbulence. In the inertial subrange, turbulent energy is neither produced nor destroyed and the energy distribution over the length scales can be described by a Kolmogorov energy spectrum (see appendix A3).
For these scales, the puff size first ($t \ll t_s$) depends on the initial puff size at the release time and increases proportional to $t^2$ ($t^2$ regime, Eq. (4)&(5)). For larger times after the release ($t \gg t_s$), the increase of the puff size becomes independent of the initial size and increases proportional to $t^3$ ($t^3$ regime, Eq.(6)). We extended the manuscript to clarify this point.

> P6 L18: "Batchelor (1952) showed that for $t \ll t_s$ the spread of a puff, or cluster of particles, is dominated by the initial velocity differences between the particles ("ballistic regime") … "

> P6 L27: "For larger times $t \gg t_s$, the rate of change of particle separation becomes independent of the initial separation, and the spread of the puff is proportional to the Richardson-Obukhov constant $C_r$ according to the Richardson-Obukhov scaling (e.g. Monin and Yaglom, 1975) …"

Figures: Please make the figures such that they all have a consistent, and readable font size.

> Figures 11-14 have been replaced.

**Additional references**

Dimotakis, P.E., 2000. The mixing transition in turbulent flows. *Journal of Fluid Mechanics*, 409, pp.69–98.

van Heerwaarden, C.C. & Mellado, J.P., 2016. Growth and Decay of a Convective Boundary Layer over a Surface with a Constant Temperature. *Journal of the Atmospheric Sciences*, 73(5), pp.2165–2177. Available at: http://journals.ametsoc.org/doi/10.1175/JAS-D-15-0315.1.

Ouellette, N.T. et al., 2006. An experimental study of turbulent relative dispersion models. *New Journal of Physics*, 8.

Yeung, P.K., 2002. Lagrangian Investigations of Turbulence. *Ann. Rev. Fluid Mech.*, 34, pp.115–142.

Yeung, P.K., Pope, S.B. & Sawford, B.L., 2006. Reynolds number dependence of Lagrangian statistics in large numerical simulations of isotropic turbulence. *Journal of Turbulence*, 7, pp.1–12.